# Linguistic analysis of health anxiety during the COVID-19 pandemic

**Alexandra D. Peterson** [1]◕, **Mindy M. Kibbey** [2]◕, **Samantha G. Farris** [2]◕ *

**1** Graduate School of Applied and Professional Psychology, Rutgers, The State University of New Jersey, Piscataway, New Jersey, United States of America, **2** Department of Psychology, Rutgers, The State University of New Jersey, Piscataway, New Jersey, United States of America

◕ These authors contributed equally to this work.
* samantha.farris@rutgers.edu

**Data Availability Statement:** https://osf.io/kfh53.

**Funding:** The authors received no specific funding for this work.

## Abstract

Health anxiety, which is defined as fear of having or contracting serious physical illness, is particularly salient in light of the COVID-19 pandemic. We conducted a mixed methods study in which 578 narrative samples were analyzed using Linguistic Inquiry and Word Count (LIWC) software to determine linguistic markers from six LIWC categories relevant to cognitive-behavioral features of health anxiety. Broad linguistic predictors were analyzed through three backward elimination regression models in order to inform subcategory predictors of each area of health anxiety. Thus, both broad and specific linguistic predictors of general health anxiety, virus-relevant body vigilance, and fears of viral contamination were examined. Greater use of affective category words in written narratives predicted general health anxiety, as well as body vigilance and viral contamination fears. These findings represent the first direct demonstration of linguistic analysis of health anxiety and provide nuanced information about the nature and etiology of health anxiety.

## Introduction

The impact of the coronavirus disease 2019 (COVID-19) pandemic and subsequent infection control restrictions has led to elevated levels of anxiety across the global population, due to both actual (e.g., viral infection) and perceived (e.g., fear of viral infection) threats [1]. Pre-existing daily stressors coupled with the additional stressors associated with the COVID-19 pandemic (e.g., social isolation, financial instability, food insecurity) can create a compounded effect of distress [2]. While some individuals remain resilient to highly stressful events such as widespread viral outbreaks, those who respond with excessive anxiety—including individuals prone to health anxiety—are at greater risk for prolonged distress [3]. In fact, "COVID Stress Syndrome" has been proposed as a pandemic-related adjustment disorder marked by clinically significant levels of distress and functional impairment [3]. While an adjustment disorder marks development of behavioral or emotional symptoms in response to an identifiable stressor—that are out of proportion to the intensity of that stressor [4]—COVID Stress Syndrome is a recently proposed adjustment disorder specific to the SARSCoV2 infection, with identifiable domains including: fear of viral infection, excessive worry about the pandemic,

**Competing interests:** The authors have declared that no competing interests exist.

and pandemic-related traumatic stress symptoms [5]. These symptoms share characteristics of health anxiety.

Health anxiety is characterized by fears of being exposed to and contracting a serious illness, or, the belief that one has already developed a serious disease or medical condition [6]. More broadly, health anxiety can refer to the tendency to over-attend to and become alarmed by health-related stimuli [4], and ranges on a continuum from mild to severe. While mild to moderate health anxiety may be adaptive and lead to engagement in appropriate health and safety measures, people with severe health anxiety frequently exhibit high levels of functional impairment, as well as overutilization of healthcare services [7, 8]. Clinically elevated health anxiety is regularly expressed as persistent worry and maladaptive beliefs about health-related stimuli, along with excessive engagement in subsequent compensatory behaviors (e.g., excessive handwashing, reassurance-seeking). Lifetime prevalence of clinically elevated health anxiety ranges from 1.3% to 10% in the general population [4]. Studies have shown that elevated health anxiety is a transdiagnostic vulnerability factor exhibited in multiple DSM-5 psychiatric disorders such as illness anxiety disorder and somatic symptom disorder as well as panic disorder and obsessive-compulsive disorder [4, 9].

Heightened body vigilance, which is often a feature of health anxiety, is particularly relevant during widespread viral outbreaks, as individuals may misinterpret somatic stress reactions (e.g., shortness of breath, fatigue, increased muscle tension) as signs of infection [10, 11]. Increased engagement in safety seeking behaviors (SSB) is also common during acute viral events. SSB, as discussed by Salkovskis [12], refer to activities that exceed necessary safety precautions and/or that individuals engage in to feel safe—reinforcing the belief that the only reason one remained safe is due to their engagement in these unnecessary and 'false' safety behaviors. One study observed that precautionary behaviors, such as using hand sanitizer and avoiding social events, are commonly exhibited by individuals affected by the COVID-19 pandemic [13]. However, while engagement in these behaviors may be adaptive and aligned to public health recommendations, higher levels of engagement were found to be correlated with heightened levels of anxiety and stress [13], suggesting that elevated worry and psychological distress is linked to excessive engagement in SSB.

Linguistic analysis of COVID-19 distress is an emerging area of research [14, 15], and published research detailing linguistic analysis of health anxiety in the context of the COVID-19 pandemic is limited [16]. Linguistic analysis has shown that the words individuals use can indicate and even predict health behavior and mental illness. However, few studies have examined linguistic markers of anxiety. Extant findings do suggest a level of attentional self-focus associated with symptoms of anxiety, as indicated by the use of first-person singular pronouns [17, 18]. Linguistic analysis of verb tense yields information about the temporal focus of attention, which is relevant to worry and anxiety processes. According to Borkovec's [19] view, engaging in worry represents an individual's attempt to prevent hypothetical negative events from occurring, and, in turn, contributes to a perceived sense of control over the threat. Thus, people who frequently engage in worry may focus their attention—and, therefore, their word use—on the future, as opposed to in the present. Indeed, use of future-tense words has been shown to be a significant predictor of *generalized anxiety disorder* diagnosis [20]. However, no published studies to our knowledge have examined linguistic predictors specific to *health anxiety*.

Therefore, the primary aim of the current study was to assess broad linguistic predictors of general health anxiety in the context of the COVID-19 pandemic. We hypothesized that use of words from the broad LIWC categories of affective, cognitive, and perceptual word processes would predict elevated general health anxiety. Specifically, we expected that anxiety-related

words would predict elevated general health anxiety, based on previous findings in the literature [18]. For a list of selected LIWC categories and word examples, see Table 1.

In addition, we aimed to identify linguistic predictors of COVID-19-related health anxiety in terms of (a) vigilance to virus-relevant bodily sensations and (b) fears of viral contamination. We hypothesized that the broad LIWC categories of perceptual and biological word processes would predict elevated virus-relevant body vigilance. Specifically, we expected that health-related words would predict elevated virus-relevant body vigilance. In terms of viral contamination fears, we hypothesized that the broad LIWC categories of affective word processes, drives words, and time orientation words would predict greater scores on this measure. Specifically, we hypothesized that both anxiety-related and future-focus words would predict elevated fears of viral contamination.

**Table 1. LIWC categories of health anxiety.**

| Process of Health Anxiety | Example Words | Mean (*SD*) |
|---|---|---|
| **Affective processes** | happy, cried | 5.8 (1.7) |
| Positive emotion | love, nice | 2.0 (1.1) |
| Negative emotion | hurt, nasty | 3.5 (1.4) |
| *Anxiety* | worried, fearful | 1.9 (1.0) |
| *Anger* | hate, annoyed | 0.3 (0.4) |
| *Sadness* | crying, sad | 0.6 (0.6) |
| **Cognitive processes** | cause, ought | 13.6 (2.7) |
| Insight | think, know | 2.5 (1.1) |
| Causation | because, effect | 2.7 (1.2) |
| Discrepancy | should, would | 1.3 (0.8) |
| Tentative | maybe, perhaps | 3.1 (1.3) |
| Certainty | always, never | 1.7 (0.9) |
| Differentiation | hasn't, but | 3.8 (1.4) |
| **Perceptual processes** | look, heard | 1.7 (0.9) |
| See | view, saw | 0.5 (0.5) |
| Hear | listen, hearing | 0.1 (0.3) |
| Feel | feels, touch | 1.0 (0.7) |
| **Biological processes** | eat, pain | 2.7 (1.4) |
| Body | hands, spit | 0.5 (0.5) |
| Health | clinic, flu | 1.9 (1.0) |
| Sexual | love, horny | 0.01 (0.1) |
| Ingestion | dish, eat | 0.3 (0.5) |
| **Drives** | take, social | 7.9 (2.0) |
| Affiliation | ally, friend | 2.7 (1.4) |
| Achievement | win, better | 1.9 (1.1) |
| Power | superior, bully | 1.5 (.08) |
| Reward | take, benefit | 1.5 (0.9) |
| Risk | danger, doubt | 0.8 (.07) |
| **Time Orientation** | | |
| Past focus | ago, did | 3.2 (1.7) |
| Present focus | today, is | 13.8 (2.7) |
| Future focus | may, will | 1.5 (1.0) |

All reported values are measured in percentages.

## Methods

### Participants

Undergraduate students of at least 18 years of age and enrolled in at least three courses at a large state university in New Jersey completed an online study in response to the COVID-19 pandemic. The university campus closed on March 12, 2020, and data were collected between April 7 and May 9, 2020. The study was administered through Qualtrics, a secure web-based platform. The Qualtrics survey included demographic information (biological sex, age, race, ethnicity, and zip code of current residence), a narrative writing task concerning psychological distress and personal impact of the COVID-19 pandemic (described below), as well as a variety of quantitative self-report instruments, the results of which are reported elsewhere [21–23]. Data collection occurred anonymously online. Participants were provided with an IRB-approved anonymous consent form. Participants were then given the opportunity to opt-in (i.e., informed consent) to proceed with the study, which was fully digital and administered through a secure, cloud-based software (e.g., Qualtrics). All procedures were approved by the Rutgers University Institutional Review Board (Approval #PRO2020000808).

### Qualitative narrative essay

For the qualitative portion of the study, participants were provided with the following narrative writing prompt instructions:

> Please write a paragraph (10–15 sentences) describing how the recent events of the COVID-19 pandemic have caused disruption and/or distress in your daily life. As you write, do not try to censor yourself or block out distressing thoughts or feelings; just notice them as you write and allow them to enrich the details of your paragraph to help us really understand your personal experience.

Participants were also provided with several example questions to help aid in the writing response (e.g., '*What has been the most stressful or worrisome aspect of the situation for you?*'). All narratives were required to be at least 1,000 characters in length to ensure adequate descriptive detail; this requirement was unknown to participants unless they attempted to submit a response shorter than the required length, in which case they were prompted to add more detail before proceeding. Average time spent writing was 11.3 minutes (*SD* = 7.4) and average length of narrative text was 274 words (*SD* = 106).

### Quantitative measures

Three quantitative measures were used to index facets of health anxiety.

**Short Health Anxiety Inventory (SHAI).**   The SHAI [24, 25] is a 14-item measure that screens for general health anxiety in the past six months. Participants are asked to respond to each item by indicating which of four statements best describes them (e.g., 0 = *I do not worry about my health* to 3 = *I spend most of my time worrying about my health*). Scores are added for a total composite score between 0 and 42, with higher scores indicating a higher level of health anxiety. Internal consistency in the current sample was $\alpha$ = .86.

**Modified Body Vigilance Scale (BVS).**   The BVS [26] is used to assess the degree of attentional focus to bodily sensations related to anxiety in the past week, and item 4 was modified for study purposes to assess vigilance to sensations associated with viral symptoms. Participants rate each item on a scale ranging from 0 (*Not at all like me*) to 10 (*Extremely like me*).

The fourth item in the scale involves separate ratings for attention to 15 different sensations, which were adapted from the original measure to assess symptoms specifically associated with COVID-19 (i.e., cough, fever, shortness of breath, tiredness, nasal congestion, runny nose, sore throat, aches and pain, diarrhea, chest pain/discomfort, faintness, sweating/clammy hands, upset stomach, nausea, and dizziness). Based on previously established scoring recommendations [27], scores from items 1, 2, and 4 were added for a total between 0 and 30. Internal consistency in the current sample was α = .71.

**Fear of Illness and Virus Evaluation—Adult report form (FIVE).** The FIVE [28] is a 35-item self-report measure consisting of four sections. We choose to use only the fears of viral contamination subscale (Part 1) in our analysis, as we considered this scale to be more directly related to the conceptualization of health anxiety as compared to the other subscales. This subscale assesses fears associated with viral illness and contamination (e.g., *I am afraid I will get very, very sick if I catch a bad illness or virus*). Participants are asked to rate their agreement with each item on a scale ranging from 1 (*Not true for me at all, I am not afraid of this at all*) to 4 (*Definitely true, I am afraid all of the time*). Internal consistency in the current sample was α = .89 for Part 1: Fears about Contamination and Illness.

## Data analysis

**Data screening.** A total of 624 participants successfully completed the full survey and provided valid qualitative data, which was screened for abnormalities and outliers. Specifically, participants who spent greater than 1 hour on the narrative essay task were excluded from analyses, due to suspected non-compliance with task instruction (*n* = 41). In addition, cases with extreme outliers on select narrative indices (described below) were also removed (*n* = 5). Thus, the final analyzed sample included 578 cases. Narrative #680 was discovered as a duplicate of #677 post-analyses. Consent for publication of raw data was not obtained, but dataset is anonymous in a manner that can easily be verified by any user of the dataset. Publication of the dataset clearly and obviously presents minimal risk to confidentiality of study participants.

**Qualitative data processing.** Qualitative data were analyzed using LIWC 2015 software [29], which defines over 74 different categories, most of which contain several dozens or hundreds of words. Each narrative was analyzed to determine the percentage of words used from six different LIWC psychological categories relevant to health anxiety (i.e., affective, cognitive, perceptual, biological, time orientation, and drives; see Table 1 for a summary of each LIWC category and the corresponding subcategories). These percentage scores were used as predictor variables in the primary test of the study aims. Given that a high degree of redundancy can occur when including both broad categories and subcategories in the same analysis [30], the broad LIWC categories were analyzed as initial predictors to inform the selection of specific LIWC subcategories to utilize in subsequent analyses.

**Quantitative data analyses.** Data analyses were conducted using SPSS Version 23.0 (IBM). Quantitative health anxiety indices (i.e., SHAI, BVS, FIVE) were the outcome variables in all analyses. An initial set of backward elimination regression models were conducted to identify the broad LIWC category markers that were predictors of each health anxiety outcome: affective processes, cognitive processes, perceptual processes, biological processes, time orientation, and drives. These retained LIWC categories were used to inform which subcategories to examine in the subsequent prediction model (Fig 1). The selected subcategories within each significant broad category were tested in a multiple regression model. Thus, a total of three initial backward prediction and three multiple regression models were conducted to identify the predictive value of relevant LIWC processes in relation to health anxiety indices.

Measure of Health Anxiety
1. General Health Anxiety
2. Virus-Relevant Body Vigilance
3. Viral Contamination Fears

Backward Elimination Regression of
**Selected Linguistic Categories**

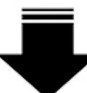

Broad Linguistic Predictors

Backward Elimination Regression of
**Broad Linguistic Predictor Subcategories**

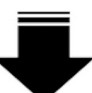

Subcategory Linguistic Predictors

**Fig 1. Backwards regression model of data analysis.**

## Results

A summary of the main findings is reflected in Table 2, as well as illustrated in Fig 2. The final sample ($N$ = 578) was predominantly female (72.7%) and had a mean age of 20.2 years ($SD$ = 2.19). Participants self-identified their race and ethnicity as white (50.9%), Asian (30.1%), Black or African American (6.7%), other (12.3%), and Hispanic or Latino (14.5%). More than a third of participants (39.1%) reported positive COVID-19 test incidence in their social network and 24.2% reported medical vulnerability to COVID-19.

Mean scores for the quantitative health anxiety measures used in this study are presented in Table 3. All three health anxiety indices were significantly inter-correlated, though not

**Table 2. Summary of findings.**

| Measure | Broad LIWC Predictors | Subcategory LIWC Predictors |
|---|---|---|
| **General Health Anxiety** | Affective Processes[a] | Anxiety[a] |
| | Cognitive Processes[b] | Differentiation[b] |
| | Drives[b] | – |
| **Virus-Relevant Body Vigilance** | Affective Processes | Anxiety |
| | Time Orientation | Future Focus[b] |
| **Fears of Viral Contamination** | Affective Processes[a] | Anxiety[a] |
| | Cognitive Processes[b] | Differentiation[b]; Certainty |
| | Perceptual Processes | – |

[a]Reflects finding consistent with *a priori* hypotheses.

[b]Reflects a low use of category words in narrative samples as a LIWC predictor of the measure.

**Fig 2. Summary of findings.**

redundant. The SHAI was moderately correlated with the BVS ($r = .573$, $p < .001$) and the FIVE ($r = .420$, $p < .001$), and the BVS with the FIVE ($r = .292$, $p < .001$). Thus, these three measures reflect related, yet distinct aspects of health anxiety.

### Regression analyses (Table 4)

**General health anxiety.** In the initial model, three LIWC categories accounted for 3.0% of variance in SHAI scores assessing general health anxiety ($F[3, 574] = 5.96$, $p < .001$), which was driven by high narrative use of affective words and low use of cognitive words and words relating to drives. In the subsequent subcategory predictor model, the LIWC predictors accounted for 7.7% of variance in the SHAI scores ($F[14, 563] = 3.34$, $p < .0001$). High use of anxiety-related words and low use of words related to differentiation were predictive of general health anxiety.

**Table 3. Descriptive statistics of quantitative study measures.**

| Variable | Mean | SD | Observed Range (Min.) | Observed Range (Max.) | Possible Range |
|---|---|---|---|---|---|
| SHAI | 12.4 | 5.8 | 1 | 39 | 0–42 |
| BVS | 12.5 | 5.6 | 0 | 27.6 | 0–30 |
| FIVE | 18.8 | 6.0 | 9 | 36 | 9–36 |

SHAI = Short Health Anxiety Inventory; BVS = Adapted Body Vigilance Scale; FIVE = Fear of Illness and Virus Evaluation, Part 1 (Fears of Viral Contamination).

**Table 4.** LIWC subcategory predictors of health anxiety, body vigilance, and fears of viral contamination.

| General Health Anxiety | | | | | | 95% CI | |
|---|---|---|---|---|---|---|---|
| LIWC Predictors | B | SE | β | t | p | LL | UL |
| Anxiety | 1.26 | 0.24 | 0.22 | 5.20 | < .001** | 0.79 | 1.74 |
| Anger | 0.84 | 0.57 | 0.06 | 1.49 | 0.138 | -0.27 | 1.96 |
| Sadness | 0.20 | 0.40 | 0.02 | 0.49 | 0.622 | -0.59 | 0.99 |
| Insight | 0.15 | 0.22 | 0.03 | 0.71 | 0.481 | -0.27 | 0.58 |
| Causation | 0.06 | 0.21 | 0.01 | 0.26 | 0.795 | -0.36 | 0.47 |
| Discrepancy | 0.47 | 0.32 | 0.07 | 1.48 | 0.139 | -0.15 | 1.09 |
| Tentative | -0.17 | 0.19 | -0.04 | -0.91 | 0.366 | -0.55 | 0.20 |
| Certainty | 0.06 | 0.26 | 0.01 | 0.22 | 0.830 | -0.46 | 0.57 |
| Differentiation | -0.49 | 0.18 | -0.12 | -2.76 | 0.006* | -0.84 | -0.14 |
| Affiliation | -0.23 | 0.18 | -0.05 | -1.29 | 0.198 | -0.58 | 0.12 |
| Achieve | -0.17 | 0.23 | -0.03 | -0.72 | 0.471 | -0.62 | 0.29 |
| Power | -0.32 | 0.29 | -0.05 | -1.10 | 0.270 | -0.89 | 0.25 |
| Reward | 0.24 | 0.28 | 0.04 | 0.87 | 0.382 | -0.30 | 0.78 |
| Risk | 0.02 | 0.37 | 0.00 | 0.06 | 0.955 | -0.70 | 0.74 |
| Virus-Relevant Body Vigilance | | | | | | 95% CI | |
| LIWC Predictors | B | SE | β | t | p | LL | UL |
| Anxiety | 0.64 | 0.23 | 0.12 | 2.78 | 0.006* | 0.19 | 1.10 |
| Anger | 0.73 | 0.54 | 0.06 | 1.34 | 0.180 | -0.34 | 1.80 |
| Sadness | 0.02 | 0.38 | 0.00 | 0.05 | 0.958 | -0.73 | 0.77 |
| Future focus | -0.64 | 0.22 | -0.12 | -2.87 | 0.004* | -1.08 | -0.20 |
| Fears of Viral Contamination | | | | | | 95% CI | |
| LIWC Predictors | B | SE | β | t | p | LL | UL |
| Anxiety | 1.14 | 0.25 | 0.19 | 4.60 | < .001** | 0.65 | 1.63 |
| Anger | -0.41 | 0.58 | -0.03 | -0.71 | 0.481 | -1.55 | 0.73 |
| Sadness | 0.18 | 0.41 | 0.02 | 0.43 | 0.671 | -0.64 | 0.99 |
| See | 0.09 | 0.48 | 0.01 | 0.20 | 0.846 | -0.85 | 1.04 |
| Hear | 1.71 | 0.96 | 0.07 | 1.77 | 0.077 | -0.19 | 3.60 |
| Feel | 0.65 | 0.38 | 0.08 | 1.70 | 0.090 | -0.10 | 1.39 |
| Fears of Viral Contamination | | | | | | 95% CI | |
| LIWC Predictors | B | SE | β | t | p | LL | UL |
| Insight | -0.02 | 0.24 | 0.00 | -0.07 | 0.943 | -0.49 | 0.46 |
| Causation | -0.22 | 0.22 | -0.04 | -1.04 | 0.301 | -0.65 | 0.20 |
| Discrepancy | 0.07 | 0.32 | 0.01 | 0.20 | 0.839 | -0.56 | 0.69 |
| Tentative | 0.17 | 0.20 | 0.04 | 0.86 | 0.389 | -0.22 | 0.55 |
| Certainty | 0.70 | 0.27 | 0.11 | 2.62 | 0.009* | 0.18 | 1.22 |
| Differentiation | -0.69 | 0.18 | -0.16 | -3.79 | < .001** | -1.05 | -0.33 |

CI = confidence interval; LL = lower limit; UL = upper limit.

*p < .05.

**p < .001.

**Virus-relevant body vigilance.** In the initial model, two LIWC categories accounted for 1.9% of variance in the modified BVS scores ($F_{[2, 575]}$ = 5.66, $p$ = .004), which was driven by high use of affective words and low use of future attentional focus words. In the subsequent subcategory predictor model, the LIWC predictors accounted for 2.8% of variance in the

virus-related BVS scores ($F[4, 573] = 4.11$, $p = .003$). High use of anxiety-related words and low use of words related to future attentional focus were predictive of body vigilance.

**Fears of viral contamination.** In the initial model, three LIWC categories accounted for 2.3% of variance in the FIVE scores ($F[3, 574] = 4.58$, $p = .004$), which was driven by high use of affective and perception words, and low use of cognitive words. In the subsequent subcategory predictor model (Table 4), the LIWC predictors accounted for 9.0% of variance in the FIVE scores ($F[12, 565] = 4.68$, $p < .0001$). Specifically, high use of anxiety-related words and certainty-related words, and low use of words related to differentiation, were predictive of viral contamination fears.

## Discussion

The purpose of this study was to examine the linguistic predictors of health anxiety in the context of the COVID-19 pandemic, as well as COVID-19-related health anxiety, specific to virus-relevant body vigilance and fears of viral contamination. Both broad and specific linguistic category predictors were analyzed. The results of the present study support the hypothesis that greater general health anxiety was predicted by higher use of words related to affective processes in general, and more specifically, by higher use of anxiety subcategory words. These trends were also reflected in our findings regarding virus-relevant body vigilance and viral contamination fears, which were partially consistent with our original hypotheses.

Our findings are consistent with previous evidence illustrating high use of negative emotion words in narrative writing surrounding viral illness outbreaks, including in the COVID-19 pandemic [14, 15, 31]. Although our study was conducted through a systematic scientific survey format, other studies have examined data that is generated in an unprompted fashion, through analysis of online language platforms such as Weibo [15] and Facebook [14], as well as language used in news reports [15]. It is possible that differences in language use exist when comparing data gathered via systematic, experimental methodology versus data that is generated naturalistically. Future research could examine if differences in language use in online platforms, such as Reddit, may be attributable to participant self-selection bias.

Our results are also consistent with findings of anxious individuals using words related to anxiety [18]. However, the current study is the first to support this finding through written narratives rather than transcribed oral samples (as in Sonnenschein, et al. [18]), marking the first direct demonstration of word use related to affective processes and anxiety as predictors of elevated general health anxiety.

Counter to our original hypothesis, analyses demonstrated that lower use of words related to cognitive processes, specifically, lower use of differentiation words, were predictive of elevated general health anxiety. One explanation for these findings may be linked to the degree of cognitive complexity exhibited by anxious participants. While a higher use of cognitive processes words may indicate a higher degree of cognitive complexity [32], participants with heightened general health anxiety may not be actively differentiating, integrating, or considering solutions to COVID-19-related distress, therefore using low levels of these category words. This theory is further supported by our finding regarding low use of differentiation words as a marker for heightened general health anxiety, indicating that the less participants were making clear distinctions (as exhibited through their writing), the greater their levels of health anxiety.

### Limitations

There are at least two potential limitations concerning the results of this study. First, LIWC is unable to detect irony, idioms, sarcasm, or even the issue of multiple meanings of words [33, 34]. Another limitation of LIWC is the matter of word overlap between broad categories and

corresponding subcategories, which posed significant issues for the analysis design of the present study. Because LIWC is conceptualized in such a way that any one word may be contained within both a subcategory and a corresponding broad category, the issue of potential overlap exists when selecting word categories for target analysis [29]. The present study was ultimately designed with the intention of avoiding this issue altogether through the strategic use of backwards regression models (Fig 1). A second potential limitation of the current study concerns the low level of variance accounted for by identified linguistic markers. Significant linguistic predictors accounted for 1.9%–9.0% of variance in the health anxiety indices analyzed. Although small effect sizes are common in the context of cognitive and behavioral research [35], it is important to consider the existence of additional mediators that may account for the remaining percentage of variances in the target measures.

## Strengths and implications

This study can be seen as a first step toward integrating two lines of research, linguistic analysis and health anxiety (specifically related to COVID-19). Few studies have examined the predictive nature of linguistic features of health anxiety, or of health anxiety in the context of viral illness outbreaks. Thus, these results represent the first direct demonstration of linguistic analysis of health anxiety in the context of the COVID-19 pandemic, as well as COVID-19-related health anxiety specific to virus-relevant body vigilance and fears of viral contamination.

Our results reveal that general health anxiety as well as virus-relevant body vigilance and fears of viral contamination are each driven by high use of emotion-focused language in expressive writing—important not only to the conceptualization of health anxiety experienced as a result of the COVID-19 pandemic, but also to understanding components of health anxiety specific to COVID-19 distress.

We would also like to highlight the data-driven methodological approach used in our analysis of linguistic predictors. This method was carefully formulated in order to avoid the matter of LIWC category overlap and risk for score inflation, as noted in the Limitations section of this paper. This approach increased the construct validity of LIWC as a reliable measure of word use in participant narratives, which may prove useful for future linguistic analyses.

## Author Contributions

**Conceptualization:** Alexandra D. Peterson, Mindy M. Kibbey, Samantha G. Farris.

**Data curation:** Alexandra D. Peterson.

**Formal analysis:** Alexandra D. Peterson, Samantha G. Farris.

**Investigation:** Alexandra D. Peterson.

**Methodology:** Alexandra D. Peterson, Samantha G. Farris.

**Project administration:** Samantha G. Farris.

**Supervision:** Mindy M. Kibbey, Samantha G. Farris.

**Writing – original draft:** Alexandra D. Peterson, Mindy M. Kibbey.

**Writing – review & editing:** Alexandra D. Peterson, Mindy M. Kibbey, Samantha G. Farris.

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
