## [Decision Letter · Decision Letter 0]

2 Nov 2023

PONE-D-23-27003Linguistic analysis of health anxiety during the COVID-19 pandemicPLOS ONE

Dear Dr. Farris, Thank you for submitting your manuscript to PLOS ONE. After careful consideration, we feel that it has merit but does not fully meet PLOS ONE’s publication criteria as it currently stands. Therefore, we invite you to submit a revised version of the manuscript that addresses the points raised during the review process.

We look forward to receiving your revised manuscript.

Kind regards,

Syed Hassan Ahmed

Guest Editor

PLOS ONE

Journal Requirements:

**Additional Editor Comments:**

Kindly adjust manuscript formatting in accordance with the journal's guidelines.

Reviewers' comments:

Reviewer's Responses to Questions

**Comments to the Author**

1. Is the manuscript technically sound, and do the data support the conclusions?

Reviewer #1: Yes

Reviewer #2: Yes

2. Has the statistical analysis been performed appropriately and rigorously? 

Reviewer #1: Yes

Reviewer #2: Yes

3. Have the authors made all data underlying the findings in their manuscript fully available?

Reviewer #1: Yes

Reviewer #2: Yes

4. Is the manuscript presented in an intelligible fashion and written in standard English?

Reviewer #1: Yes

Reviewer #2: Yes

5. Review Comments to the Author

Reviewer #1: The authors should consider the following.

1. definition of health anxiety (how is it different from anxiety disorder?)

2. definition of covid stress syndrome and adjustment disorder

3. Every paragraph should be starting with ''topic sentence or words''.

4. the authors should consider explaining ''distress vs normal stress'' and the differences in bodily reaction amongst individuals. Exploring other risk factors to distress will also be great.

Reviewer #2: 1. The authors mention excluding "small/limited" responses. It would be helpful, if they can mention that what no of words or characters were deemed as a small response from the participants.

2. The discussion section could highlight studies that covered platform like Reddits (if available for similar domain) and can draw a comparison with your own study as this can also give an insight to future researchers as to which method is more objective or could be explored more. Personal surveys or already posted comments.

6. PLOS authors have the option to publish the peer review history of their article (what does this mean?). If published, this will include your full peer review and any attached files.

Reviewer #1: No

Reviewer #2: **Yes: **Summaiyya Waseem

---

## [Author Response · Author response to Decision Letter 0]

22 Dec 2023

Manuscript Number: PONE-D-23-27003

Title: Linguistic analysis of health anxiety during the COVID-19 pandemic

1. Please ensure that your manuscript meets PLOS ONE's style requirements, including those for file naming. The PLOS ONE style templates can be found at https://journals.plos.org/plosone/s/file?id=wjVg/PLOSOne_formatting_sample_main_body.pdf and https://journals.plos.org/plosone/s/file?id=ba62/PLOSOne_formatting_sample_title_authors_affiliations.pdf.

RESPONSE: We have reviewed the PLOS ONE style templates at the links provided and updated the manuscript to meet the specified requirements, which can be viewed via the “track changes” markup.

RESPONSE: We have received approval from our Institutional Review Board that our consent form and protocol as written allow us to share the de-identified study data. Data was uploaded to the Open Science Framework (OSF) repository, and is available here: https://osf.io/kfh53

a. If there are ethical or legal restrictions on sharing a de-identified data set, please explain them in detail (e.g., data contain potentially sensitive information, data are owned by a third-party organization, etc.) and who has imposed them (e.g., an ethics committee). Please also provide contact information for a data access committee, ethics committee, or other institutional body to which data requests may be sent.

RESPONSE: We have received approval from our Institutional Review Board that our consent form and protocol as written allow us to share the de-identified study data. Thus, we have removed the language from our paper (line 191): The archived dataset is available from the corresponding author upon reasonable request. Although the collection of personally identifiable information was limited, and no data that can be used to readily identify participants was collected, the researchers do not wish to release the data publicly due to the qualitative nature of participant responses. We replaced this language with the following language: Consent for publication of raw data was not obtained, but dataset is anonymous in a manner that can easily be verified by any user of the dataset. Publication of the dataset clearly and obviously presents minimal risk to confidentiality of study participants.

b. If there are no restrictions, please upload the minimal anonymized data set necessary to replicate your study findings as either Supporting Information files or to a stable, public repository and provide us with the relevant URLs, DOIs, or accession numbers. For a list of acceptable repositories, please see http://journals.plos.org/plosone/s/data-availability#loc-recommended-repositories. We will update your Data Availability statement on your behalf to reflect the information you provide.

RESPONSE: Data was uploaded to the Open Science Framework (OSF) repository, and is available here: https://osf.io/kfh53

RESPONSE: We thank the reviewers for noting discrepancy of references cited. A review of the manuscript was conducted to update the reference list. A review of the reference list was then conducted, and no articles were found to be of retracted status. All references are fully cited in the reference list. 

Reviewer #1: 

1. The authors should consider the following: definition of health anxiety (how is it different from anxiety disorder?)

RESPONSE: We have added clarification on the difference between health anxiety and anxiety disorders through the addition of another paragraph. The revised text reads (line 63) as follows: Health anxiety is characterized by fears of being exposed to and contracting a serious illness, or, the belief that one has already developed a serious disease or medical condition [6]. More broadly, health anxiety can refer to the tendency to over-attend to and become alarmed by health-related stimuli [4], and ranges on a continuum from mild to severe. While mild to moderate health anxiety may be adaptive and lead to engagement in appropriate health and safety measures, people with severe health anxiety frequently exhibit high levels of functional impairment, as well as overutilization of healthcare services [7, 8]. Clinically elevated health anxiety is regularly expressed as persistent worry and maladaptive beliefs about health-related stimuli, along with excessive engagement in subsequent compensatory behaviors (e.g., excessive handwashing, reassurance-seeking). Lifetime prevalence of clinical health anxiety ranges from 1.3% to 10% in the general population, and is exhibited in psychiatric disorders such as illness anxiety disorder and somatic symptom disorder [4].

2. The authors should consider the following: definition of covid stress syndrome and adjustment disorder

RESPONSE: We have added clarification on the definitions of covid stress syndrome and adjustment disorder. The revised text reads (line 57) as follows: While an adjustment disorder marks development of behavioral or emotional symptoms in response to an identifiable stressor—that are out of proportion to the intensity of that stressor [4]—COVID Stress Syndrome is a recently proposed adjustment disorder specific to the SARSCoV2 infection, with identifiable domains including: fear of viral infection, excessive worry about the pandemic, and pandemic-related traumatic stress symptoms [5]. These symptoms share characteristics of health anxiety.

3. The authors should consider the following: Every paragraph should be starting with ''topic sentence or words''.

RESPONSE: We added a topic sentence at line 67 to provide clarity to readers. The revised text reads as follows: Health anxiety is characterized by fears of being exposed to and contracting a serious illness, or, the belief that one has already developed a serious disease or medical condition.

4. The authors should consider the following: the authors should consider explaining ''distress vs normal stress'' and the differences in bodily reaction amongst individuals. Exploring other risk factors to distress will also be great.

RESPONSE: We added the following sentence to line 44 to add clarity to “distress” out of proportion to the norm, as well as to inform risk factors of such distress: Pre-existing daily stressors coupled with the additional stressors associated with the COVID-19 pandemic (e.g., social isolation, financial instability, food insecurity) can create a compounded effect of distress.

Reviewer #2: 

1. The authors mention excluding "small/limited" responses. It would be helpful, if they can mention that what no of words or characters were deemed as a small response from the participants.

RESPONSE: Please see lines 172–178, where we define the required criteria for a complete response.

2. The discussion section could highlight studies that covered platform like Reddits (if available for similar domain) and can draw a comparison with your own study as this can also give an insight to future researchers as to which method is more objective or could be explored more. Personal surveys or already posted comments.

RESPONSE: We added the following language to address the topic of language use in other types of platforms, as well as how this may be address in future research (lines 306–312): Although our study was conducted through a systematic scientific survey format, other studies have examined data that is generated in an unprompted fashion, through analysis of online language platforms such as Weibo [14] and Facebook [13], as well as language used in news reports [14]. It is possible that differences in language use exist when comparing data gathered via systematic, experimental methodology versus data that is generated naturalistically. Future research could examine if differences in language use in online platforms, such as Reddit, may be attributed to participant self-selection bias.

---

## [Decision Letter · Decision Letter 1]

18 Jan 2024

PONE-D-23-27003R1Linguistic analysis of health anxiety during the COVID-19 pandemicPLOS ONE

Dear Dr. Farris,

Thank you for submitting your manuscript to PLOS ONE. After careful consideration, we feel that it has merit but does not fully meet PLOS ONE’s publication criteria as it currently stands. Therefore, we invite you to submit a revised version of the manuscript that addresses the points raised during the review process.

We look forward to receiving your revised manuscript.

Kind regards,

Syed Hassan Ahmed

Guest Editor

PLOS ONE

Journal Requirements:

Reviewers' comments:

Reviewer's Responses to Questions

**Comments to the Author**

1. If the authors have adequately addressed your comments raised in a previous round of review and you feel that this manuscript is now acceptable for publication, you may indicate that here to bypass the “Comments to the Author” section, enter your conflict of interest statement in the “Confidential to Editor” section, and submit your "Accept" recommendation.

Reviewer #1: All comments have been addressed

Reviewer #2: All comments have been addressed

2. Is the manuscript technically sound, and do the data support the conclusions?

Reviewer #1: Yes

Reviewer #2: Yes

3. Has the statistical analysis been performed appropriately and rigorously? 

Reviewer #1: I Don't Know

Reviewer #2: Yes

4. Have the authors made all data underlying the findings in their manuscript fully available?

Reviewer #1: Yes

Reviewer #2: Yes

5. Is the manuscript presented in an intelligible fashion and written in standard English?

Reviewer #1: Yes

Reviewer #2: Yes

6. Review Comments to the Author

Reviewer #1: The use of ''health anxiety'' is still somehow confusing and most professionals trained in mental health might still not be able to understand it. Professionals in mental health and psychiatry communicate either by using Diagnostic and statistical manual (DSM) or the international classification of diseases (ICD). According to the authors definitions of heath anxiety, it might qualify as hypochondriasis or illness anxiety disorder (ICD and DSM 5 respectively). The authors should consider adopting the DSM/ICD nomenclature or adding text to explain the equivalent of health anxiety in ICD or DSM.

Reviewer #2: (No Response)

7. PLOS authors have the option to publish the peer review history of their article (what does this mean?). If published, this will include your full peer review and any attached files.

Reviewer #1: **Yes: **DR ALEX ZUMAZUMA (MBBS, MMED PSYCHIATRY)

Reviewer #2: **Yes: **Summaiyya Waseem

---

## [Author Response · Author response to Decision Letter 1]

5 Feb 2024

Reviewer #1: 

1. The use of ''health anxiety'' is still somehow confusing and most professionals trained in mental health might still not be able to understand it. Professionals in mental health and psychiatry communicate either by using Diagnostic and statistical manual (DSM) or the international classification of diseases (ICD). According to the authors definitions of heath anxiety, it might qualify as hypochondriasis or illness anxiety disorder (ICD and DSM 5 respectively). The authors should consider adopting the DSM/ICD nomenclature or adding text to explain the equivalent of health anxiety in ICD or DSM.

RESPONSE: We have added clarification on the definition of health anxiety as a transdiasgnostic vulnerability factor exhibited in multiple DSM-5 diagnoses. The revised text reads (line 62) as follows: Clinically elevated health anxiety is regularly expressed as persistent worry and maladaptive beliefs about health-related stimuli, along with excessive engagement in subsequent compensatory behaviors (e.g., excessive handwashing, reassurance-seeking). Lifetime prevalence of clinically elevated health anxiety ranges from 1.3% to 10% in the general population [4]. Studies have shown that elevated health anxiety is a transdiasgnostic vulnerability factor exhibited in multiple DSM-5 psychiatric disorders such as illness anxiety disorder and somatic symptom disorder as well as panic disorder and obsessive-compulsive disorder [4, 5].

---

## [Editor Report · Decision Letter 2]

12 Feb 2024

Linguistic analysis of health anxiety during the COVID-19 pandemic

PONE-D-23-27003R2

Dear Dr. Farris,

We’re pleased to inform you that your manuscript has been judged scientifically suitable for publication and will be formally accepted for publication once it meets all outstanding technical requirements.

Kind regards,

Syed Hassan Ahmed

Guest Editor

PLOS ONE
---

## [Editor Report · Acceptance letter]

16 Feb 2024

PONE-D-23-27003R2 

PLOS ONE

Dear Dr. Farris, 

I'm pleased to inform you that your manuscript has been deemed suitable for publication in PLOS ONE. Congratulations! Your manuscript is now being handed over to our production team.

Kind regards, 

on behalf of

Dr. Syed Hassan Ahmed 

Guest Editor

PLOS ONE